# Productivity from a Metapragmatic Perspective: Measuring the Diachronic Coverage of the Low Level Lexico-Grammatical Construction Have the N (Body Part/Attitude) to ↔<Metapragmatic Comment> Using the COHA

Chris A. Smith

CRISCO EA4255, Université de Caen, 14000 Caen, France; chris.smith@unicaen.fr

**Abstract:** This paper seeks to address the relation between semantics, pragmatics and the productivity of a low level lexico-grammatical construction, HAVE THE N (BODY PART/ATTITUDE) TO ↔METAPRAGMATIC COMMENT. The question posed is how semantics affects productivity, in the generative sense of extensibility of a construction (a form meaning pairing). The method identifies the specificity and variations of the HAVE THE N (BODY PART/ATTITUDE) TO ↔METAPRAGMATIC COMMENT construction within the pragmeme of politeness using the COHA. Hereafter, we consider how to measure the extensibility within the onomasiological frame based on the available pool of forms expressing an attitude/emotion, i.e., the coverage or attractivity of the HAVE THE N TO construction. The paper discusses the findings, namely, how to overcome methodological issues relating to a qualitative rather than quantitative approach to the constructional architecture and the relative productivity of constructions. The experimental small scale corpus study of Have the N to in the COHA suggests that a global view of constructional architecture at multiple levels should be pertinent to identifying the extensibility potential of the construction.

**Keywords:** grammaticalization; constructional change; productivity; coverage; diachronic; distributional semantics; COHA

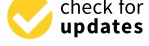



## 1. Introduction

In this paper, for the special issue "Grammaticalization across languages, levels and frameworks", we look at constructional change rather than grammaticalization, which is, strictly speaking, "defined as the development from lexical to grammatical forms and from grammatical to even more grammatical forms" (Heine and Kuteva 2002, p. 2). "Since the development of grammatical forms is not independent of the constructions to which they belong, the study of grammaticalization is also concerned with constructions and with even larger discourse segments" (Heine and Kuteva 2002, p. 2). Thus, we take the constructional network approach (Traugott and Trousdale 2013) at its word in a maximalistic view as being a holistic (all-encompassing) apparatus following Hilpert (2013) and Goldberg (2019). As Hilpert (2013, p. 9) puts it, "[c]onstructional change is more encompassing than the changes that characterize grammaticalization", which, arguably, is characterized by paradigmatization. Indeed, Hilpert (2013) observes that "[c]onstructions that lack paradigmatic oppositions, that is, forms such as the way-construction, *let alone*, or *the Xer the Yer*, would in this view fall outside the scope of grammar."

If we take a global view of the lexicogrammatical continuum and view language change as gradual and transitional, then changes involving lexis may be connected to changes involving grammatical aspects of language. If we also accept that constructions form a network of interrelated constructions of a higher and lower level as outlined in Traugott and Trousdale (2013), then lower-level constructions may have the ability to affect higher level constructions, as well as vice versa (see Hilpert 2013; Traugott and Trousdale

2013 on micro, meso and macro constructions). Given these theoretical assumptions, this paper addresses the issue of constructional change and how to determine the emergence of constructional change, as well as the productivity of such changes. Although constructional approaches are gaining ground, maybe even bolstered by this increasing popularity, the definition and properties of Cx continue to be hotly debated (see Goldberg 2019). There are also notably (and paradoxically) an increasing number of subtypes of construction grammar, such as Cx morphology or CxM (defined by Masini and Audring 2019, p. 372) as "a sign-based theory of morphology whose building blocks are constructions"), or multimodal construction grammar (Imo 2015).

Constructions[1] are defined generally as form-meaning pairings that vary in complexity and schematicity. Traugott and Trousdale (2013, p. 163) cite properties as schematicity, compositionality and productivity. Perek (2020) lists schematicity and productivity, whereas Bybee (2010, pp. 220–21) adds autonomy, prototype effects and analysability. A lexical schema is productive if it is extensible, but not necessarily open ended, as argued by Barðdal (2008, p. 1). The extension of the schema results from local analogies operating on the onomasiological (or conceptual) plane according to Traugott (2019, p. 131). Larger level constructions have been studied as being more productive (*X your way, the Xer the Yer*) which are consistent with the top–down approach usually carried out regarding the constructions.

The issues of variation and innovation (Hilpert 2013[2]; Goldberg 2016, 2019; Hoffmann 2020), or capacity for growth or extension of constructions, are currently being investigated from different perspectives including lexical/morphological perspectives (Audring et al. 2017; Booij and Audring 2018; Masini and Audring 2019). The objective is to avoid disregarding the importance of the lower-level lexicon which can often be overlooked or oversimplified and to account for the Cx architecture from a low level to a higher level. We believe there is a gap in the research on constructions and productivity that will benefit from a bottom-up approach to constructions, thus focusing on the lower-level lexical stratum. A bottom-up approach to constructions tends to consider the importance of low-level constructions and how they affect higher level constructions (Gyselinck 2020; Budts and Petré 2020).

The specificity and originality of our position is threefold: (1) starting with a lower-level construction belonging to the lexicogrammatical layer (Hilpert 2013, p. 202); (2) taking a qualitative approach to productivity and (3) relying on a communicative approach to constructions.

(1) The first aspect positions this line of research firmly to use Hilpert's (2013, p. 202) words, in the camp of "practitioners of Construction Grammar [ . . . ] who view low-level schemas as central to the description of linguistic knowledge" as opposed to the camp of those who "aim for abstract generalizations". It is fundamentally a data-driven method that allows for "acknowledging the rich networks of lexical schemas and collocations [that] characterize the actual usage of grammatical constructions" (Hilpert 2013, p. 202). It should be possible to build the architecture from the bottom as well as from the top if we assume that motivational ties are multidirectional in the constructional network.

(2) Taking things beyond frequency. It is also true that much productivity research has focused on quantitative frequency measures (Hilpert 2013), but focusing on onomasiological qualitative measures has been suggested to be a way forward (Zenner et al. 2014; Geeraerts 2016; Durkin 2016; Fernandez-Domínguez 2019; Petré 2019; Goldberg 2019; Lorenz 2020), correcting the bias in favor of more frequent abstract grammatical structures and to the detriment of less frequent lower-level lexicogrammatical structures.

(3) Using pragmatics as an entry point for the onomasiological perspective. To test the onomasiological perspective, we follow Hilpert and Bourgeois' (2020) main assumption that a pragmatic context motivates a set of constructions which in turn can be seen to fall into a larger structure, a metapragmatic construction so to speak. "It is a basic tenet of usage-based construction grammar (Goldberg 2006; Bybee 2010) that long-term linguistic changes originate from processes that are at work in actual com-

municative situations. [ . . . ] So far, however, relatively little work on constructional change addresses either the dialogical nature of language or the social context in which a particular construction is used Pl" (Hilpert and Bourgeois 2020, pp. 97–98).

Since points (2) and (3) are interrelated, we will develop them together. As Geeraerts (2016, p. 158) argues, experiential frequency, conceptual frequency and lexical frequency should not be confused as a single effect: "The frequency of occurrence of a linguistic construct results from these three types of frequency at the same time, and talking of entrenchment as a single, unitary phenomenon obscures those differences". Petré (2019, p. 161) also argues that onomasiological space is the key to measuring entrenchment and that "the emergence of a construction is better measured by a more fine-grained quantitative analysis of the contexts in which the pre-construction material occurs".

In other words, the onomasiological approach is valuable if we want to build an accurate picture of the constructicon, the global architecture of constructions, i.e., the identification of constructional families (De Smet et al. (2018, p. 205). As Goldberg (2019, p. 36) puts it, "our knowledge of language forms a construct-i-con, which includes words, partially filled words (aka morphemes), and representations that are larger than single words, all represented in a complex dynamic network, much as we have long known to be true of the lexicon.) and how constructions share semantic or grammatical space." The function of interconnecting schemas and constructions in the constructicon is to represent the multiple motivations at work in the lexico-grammatical continuum as underlined in De Smet et al. (2018, p. 206), Audring et al. (2017) and Booij and Audring (2018).

This paper seeks to bridge the gap between the pragmatic–semantic interface and constructional productivity (see Smith 2021), and takes a maximalist lexicalist view of the constructicon. The pragmatic–semantic interface is operationalized using Mey's ([1993] 2001) concept of the pragmeme. A pragmeme is defined in Mey ([1993] 2001, p. 208) as an instantiated pragmatic act, or a "generalised pragmatic act" (Mey 2010, p. 2884), examples of which are "as found in invitations, bribes, co-optations, incitements, and so on—all depending on the situation through which they are defined" (Mey 2010, p. 2884). We propose to center our analysis on the pragmeme of insolence (or impoliteness), i.e., generalized speech acts belonging to a socio-cultural situation of face-threatening or face-preserving (see Culpeper 2011; Bousfield 2008). Our specific focus will be on the metapragmatic reaction to perceived insolence via the expression *have the N to.* The objective for this paper is to identify the specificity and variations of the Have the N to ↔attitude construction within the pragmeme of insolence using the COHA or *Corpus of Historical American English*.

Our driving question is how to measure attractivity of the construction based on the extensibility within the onomasiological frame, that is the available pool of forms expressing an attitude/emotion. The assumption behind this is that productivity should be relative to the extensibility potential of a construction. Using frequency measures exclusively tends to disregard less abstract lower level constructions, and might misrepresent their actual productivity. It is notable that Goldberg (2019) has since criticized her own reference to constructions requiring "sufficient frequency" (Goldberg 2006, p. 5) as "nonsensical", reinforcing the idea that if there is a consensus that constructions exist at multiple levels of complexity and abstractness, then it is essential that the productivity of such constructions be assessed within their own relative space. The aim is to track the emergence of the constructional architecture based on the instantiations of the construction, taking into account horizontal links (synonymy, polysemy, semantic shift) between fillers. As Wray (2017, p. 2) underlines, "[ . . . ] there is more to *usage* than just patterns of frequency. We must understand what motivates the usage, and that entails recognizing the functions that they have: both cognitive and social."

Our proposal is to carry out a lexicalist bottom-up approach to Cx starting with a low-level construction and our method will rely on a fine grained distributional semantic analysis using the COHA which we will now develop.

## 2. Materials and Methods

This paper will focus on the lower-level micro construction HAVE THE N TO, a construction (or collostruction) of metapragmatic nature that takes a selection of nouns relating to body parts or nouns referring to emotion concepts. It is a low-level verbal lexico-grammatical structure with a single variable N consisting in a predicate structure around the verb HAVE + complement NP carrying a definite article THE and followed by infinitival structure TO. The construction falls under a general semantic abstract schema of metaphor and metonymy relating to body parts and emotions and attitudes (confidence, arrogance). The construction also belongs to the transitive ACS (argument structure construction) including an infinitival object complement.

In our conceptual space or onomasiological space perspective, we propose that this construction is a manifestation of what can be called the "pragmeme of politeness (approval/disapproval)" (see Allan 2016 on the pragmeme of insult). The pragmeme is a pragmatic routine as per Mey ([1993] 2001). Insolence (or an insolent speech act) and indignation (the reaction to a perceived insolent act) function as a pragmatic routine, i.e., a pragmeme, which in all likelihoods has a multimodal nature as insolence and indignation manifest in a broader communicative situation. Insolence is therefore more than an onomasiological field but rather a multimodal pragmeme. "The pragmeme is thus the embodied realization of all the pragmatic acts (or 'allopracts') that can be subsumed under it, such as the various manifestations of expressing gratitude . . . " Mey ([1993] 2001, p. 139).

We propose that the pragmeme can be operationalized as a higher order pragmatic construction. We believe that both the level of generalization of the pragmeme and its relevance in dialogic and communicative contexts give the pragmeme its potential to be a powerful entry point into the constructional architecture. The pragmeme can be seen as a (multimodal) construction (meaning form pairing) with variables; it is a higher-level abstract construction since it can be instantiated by subconstructions. Indeed, the insolence/impoliteness pragmeme could subsume a number of constructions expressing metalinguistic reactions to perceived acts of insolence or face-threats. Within this pragmeme of insolence (or impoliteness), we can assume the existence of a whole network of interconnected subconstructions, including *have the N to*, but also *What the N*, *don't give me N* and *don't get Ad with me*, to name a few. Additionally, it should be noted that in terms of grammaticalization potential, insolence or the larger pragmeme which subsumes insolence or impoliteness is fundamentally intersubjective in nature (Ghesquière et al. 2015; Traugott 2015). If intersubjectivity is an orientation toward the co-speaker or the act of speech itself, markers of intersubjectivity have been found to be compatible with right periphery usage, in turn taking positions. *<HAVE the N to>* is a metapragmatic evaluation of the co-speakers' behavior.

In this paper, we seek to test the proportional or relative space that individual micro-constructions take in the onomasiological field of insolence or metapragmatic attitude operationalized by the pragmeme. Previous work (Smith 2021) showed that the semantic extension of body part nouns (*cheek*, *lip*, *face*, *nerve*) associated with insolence is related to the emergence of *have the N to*, which then generated the lexical schema Ny (*cheeky*, *lippy*, *facy*, *nervy*) adjective associated with the sense 'insolent'. The working hypothesis is that these two low level constructions have emerged diachronically to associate body parts with insolence. This relative space corresponds to what Goldberg (2019, p. 51) has termed "coverage": "the meaning and distribution of words, combinations of words, and constructions rely on the nature of our memory. In particular, partially abstracted (lossy) structured exemplars dynamically cluster within our hyper- dimensional conceptual space. [ . . . ] new expressions are licensed to the extent that they comfortably fit within an existing cluster."

To test this coverage (or extensibility), this paper carries out a corpus-driven analysis of the variation of fillers in the N slot of the micro construction *Have the N to* using the diachronic balanced corpus COHA (Davies 2010) which has successfully been used for diachronic tracking studies (Hilpert 2013). The working hypothesis is the following: The

compared frequency of the lexically filled structures provides a measure of the representativeness of the exemplar within the field. From a cognitive perspective, it is supposed that the lexicon contains certain "driving" elements belonging to the core lexicon which have a higher entrenchment in the mental lexicon within a conceptual domain (Killgarriff et al.'s 2004 "dominant senses"). Although the relation between general frequency and representativeness is not straightforward, frequency measures and collocations scores can help to identify these potential drivers within the semantic space via consistency of meaning. Extensions of the pattern over time will provide evidence of the productivity of the pattern and potential evidence of semantico-pragmatic change. The method used is a corpus-driven collocational and syntactic analysis of the *have the N to* collostruction (lexicogrammatical pattern) using in the COHA. Collocational behavior and lexicogrammatical patterns have been shown to be instrumental for identifying patterns of usage (Gries and Stefanowitch 2004; Perek 2020). From a diachronic perspective, tracking these behaviors over time for the period 1820 to 2010 will provide some data regarding the extensibility of the pattern, and hence evidence of the onomasiological productivity of the low-level lexico grammatical pattern. From there, we discuss how to assess the representativeness of individual fillers, using the collocate analysis to assess the proportion of usage corresponding to the insolence pragmeme.

## 3. Results Tracking Noun Fillers

In this section, we track the variation in noun fillers over time using the COHA to provide frequencies (overall and diachronic) and collocate profiles of specific *have the N to* constructions.

### 3.1. The Categories of Noun Fillers

First, we look at the variation in N fillers occupying the complement slot in the verb structure *Have the N to*. The results for the query concerning N fillers in the structure *Have the N to* in the COHA are reproduced in Table 1. The query HAVE THE N TO in capitals allowed for all variations of the lemma *have* to be included: *-had*, *have*, *has*. The following table is not exhaustive as not all forms appear; frequencies below 30 do not appear as we shall see later.

**Table 1.** List of N fillers for *Have the N to* in COHA and raw frequencies.

| Have the N (Concept) | | Have the N (Body Part) | |
|---|---|---|---|
| Noun | Tokens | Noun | Tokens |
| RIGHT | 3062 | NERVE | 321 |
| POWER | 1746 | HEART | 316 |
| COURAGE | 1148 | GUTS | 196 |
| OPPORTUNITY | 646 | FACE | 35 |
| HONOR | 534 | GALL | 40 |
| ABILITY | 510 | | |
| CHANCE | 483 | | |
| MISFORTUNE | 283 | | |
| TIME | 267 | | |
| STRENGTH | 253 | | |
| MONEY | 252 | | |
| AUTHORITY | 247 | | |
| CAPACITY | 188 | | |
| SENSE | 162 | | |
| AUDACITY | 161 | | |
| GOODNESS | 150 | | |
| MEANS | 141 | | |
| GRACE | 87 | | |
| RESOURCES | 87 | | |
| FORESIGHT | 73 | | |
| TEMERITY | 72 | | |

**Table 1.** *Cont.*

| Have the N (Concept) | | Have the N (Body Part) |
|---|---|---|
| KEY | 71 | |
| ENERGY | 65 | |
| HAPPINESS | 62 | |
| DECENCY | 55 | |
| KINDNESS | 47 | |
| IMPUDENCE | 46 | |
| EFFECT | 45 | |
| LUCK | 44 | |
| WIT | 40 | |
| PLEASURE | 36 | |
| CURIOSITY | 35 | |
| SATISFACTION | 34 | |
| PATIENCE | 33 | |
| URGE | 33 | |
| WILL | 31 | |

The N fillers in Table 1 are listed from the highest to lowest raw frequency in the COHA. The left column represents concept nouns (*power*, *opportunity*, *curiosity*), whereas the right column represents body part nouns.

In order to attempt a classification of types of N fillers, the concept nouns are classified conceptually in terms of the types of evaluation they describe, thus providing a pattern of possible semantic traits of the construction. We know the construction can have literal or figurative interpretations and we know the construction is associated with a metapragmatic commentary on a situation. The nature of the commentary can vary from factual to positive or negative evaluation. From the list of nouns provided, we have broadly identified five types based on a lexical semantic analysis.

(1) Modal evaluation of right or ability: *have the right, power, authority to, capacity, strength, money, key*
(2) Evaluation of character: *decency, kindness, heart, sense, patience, honor*
(3) Factual: *chance, opportunity, time*
(4) Negative evaluation of attitude: *the impudence, gall, nerve, cheek, face, audacity temerity*
(5) Positive evaluation of attitude: *courage, foresight*

We propose that these instantiations find themselves on a politeness–impoliteness spectrum with varying degrees of judgement or evaluation (from approval to disapproval and indignation).

Whereas concept noun fillers are many, the body part fillers are far less varied. The body parts listed mostly have a metaphorical sense of impoliteness: *nerve, heart, guts, face.* Some body part nouns however are not listed (*stomach* or *cheek*) in this search but do occur, such as *cheek*, as their frequency falls below the threshold of 30; *Have the cheek* has 18 tokens, *have the stomach* 15 tokens. These frequencies are well below those of *heart, nerve, guts* which are above 30, with *nerve* and *heart* in the lead.

A further issue with the search query HAVE THE N TO is that the COHA may not reliably recognize all N slot fillers as being nouns. For instance, a search for *have the wherewithal to* produces 48 hits, which should mean these results should have been included in the previous table. However, *wherewithal* is not readily labelled as a noun, and therefore does not appear in the results. The Oxford English Dictionary (OED for hereon) labels *wherewithal* [1535] as an adverb and noun with little information regarding its formation as a compound of *where + withal*. The COHA occurrences show that *Have the wherewithal to* remains in use with a low frequency from 1840s to 2010s. The relative frequency shown in Figure 1 indicates that this is on the rise again between 2000–2010 after several periods of increased and decreased usage.

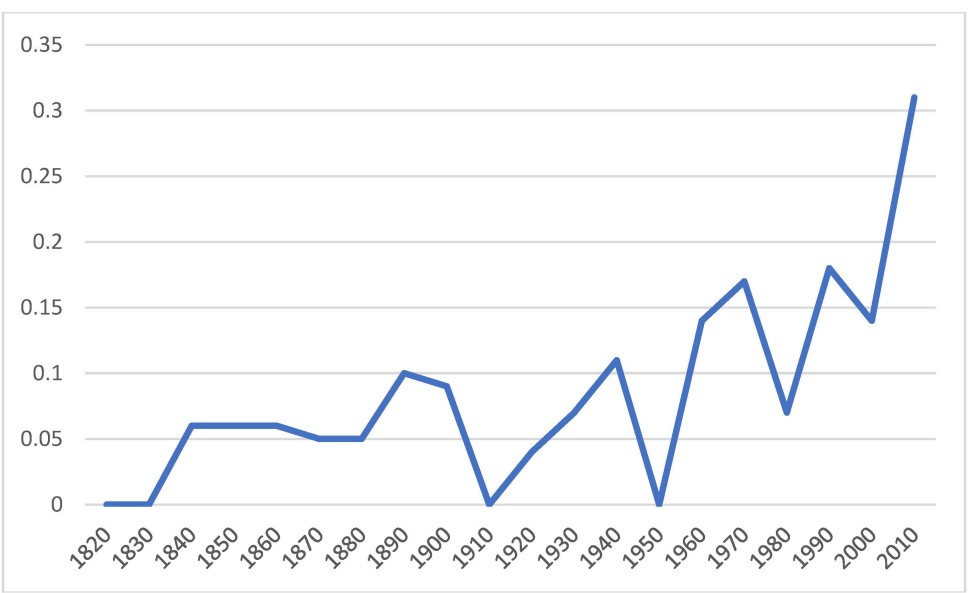

**Figure 1.** Relative frequency of *have the wherewithal to per decade* in COHA.

The meaning of *wherewithal* fits in with the conceptual category ability/capacity/but also means/resources/sense. The total number of occurrences of *wherewithal* average 362 with 48 uses in the pattern *have the N to*. Nearly all occurrences find *wherewithal* followed by *to* or *for*. Occasionally, the determiner *the* is omitted in some earlier occurrences as in (1). Note that the occurrences of N fillers in the corpus extract is highlighted in bold character for added visibility. The source of the COHA extract is provided as it appears in COHA (date, title and author).

(1) "you are the real sovereigns of Castile, enjoying all the rights and revenues of royalty, while I, stripped of my patrimony, have scarcely **wherewithal** to procure the necessaries of life." (1864. *The History of the Reign of Ferdinand and Isabella the Catholic – Volume 1. Prescott*, William Hickling)

(2) Things I can't understand. I could be better educated, Mr. Keele. That's why I've come to you. I want some help. "I leaned back. If he found gold, he should have the **wherewithal** to get in there and back without my help" (1949. *Amazing stories; Valley of the Croen*, Tarbell, Lee)

(3) "Hey, what's going on here tonight?" I could not give him anything but a shrug for a reply. And try as I might, when I later kissed her goodbye at the door, I did not seem to have the **wherewithal** to feign even a little filial affection. (1974. *My life as a man*. Philip Roth)

Notwithstanding this issue of non exhaustivity of the search query, overall, as expected, since words for concepts have more variation, there are considerably more concept fillers than body part fillers (*heart*, *nerve*, *face*, *guts*, *gall*). In addition, the frequency of occurrence of the concept nouns is much higher on the whole than body part fillers, with *right*, *power* and *courage* occurring far more frequently than body noun fillers. The extensibility of the schema is limited since names for body parts do not evolve quickly (see Durkin 2016, who shows that borrowing tends to be restricted in certain conceptual domains).

The compared relative frequencies of highest ranking are shown in Figure 2.

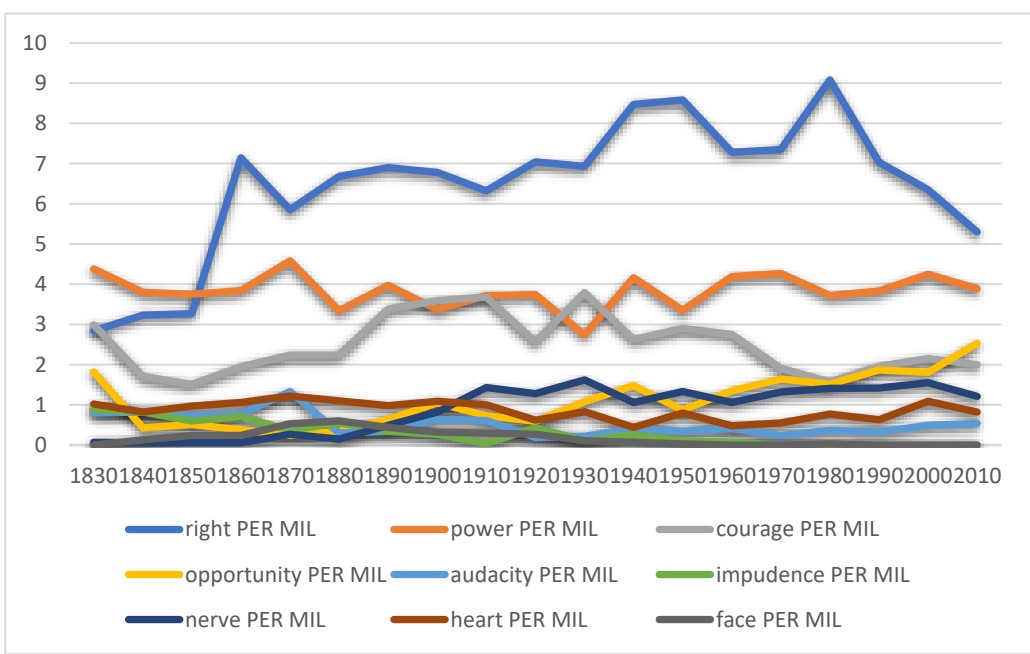

**Figure 2.** Relative frequencies for several *Have the N* 1820–2010 in COHA.

The relative frequencies in Figure 2 are indicative of stability over the 200-year period covered by the COHA, with some variations in the higher frequency N: *right*, *power* and *courage*. In the lesser frequency category, *nerve* is a clear winner with its increasing frequency (which we will return to). This suggests that the pattern is already well established by the 1800s.

### 3.2. Metapragmatic Comment Fillers

Based on the variation in concept fillers shown previously in Table 1, it also appears that a proportion fits the metapragmatic comment on the courage/insolence or approval/disapproval spectrum: namely *courage*, *strength* (on the positive spectrum), *audacity* and *temerity* (on the negative spectrum). Table 2 lists the frequency of some of these N fillers.

**Table 2.** Have the N (+/− impudence/courage) in COHA.

| Have the N | Tokens |
| :---: | :---: |
| *Courage* | 1148 |
| *Strength* | 253 |
| *Audacity* | 161 |
| *Temerity* | 72 |
| *Impudence* | 46 |

These fillers are comparable semantically to the far less frequent body part fillers shown in Table 3 carrying metonymical meanings for insolence, impudence, impoliteness and courage.

**Table 3.** Have the N (body part) in COHA.

| Have the N | Tokens |
| :---: | :---: |
| *Nerve* | 321 |
| *Heart* | 316 |
| *Guts* | 196 |
| *Face* | 25 |
| *Gall* | 40 |

The development of the politeness/impoliteness association of body parts historically appears related to the emergence of construction <*Have the N to*> insolence as proposed in Smith (2021). If we posit a micro schema 1: Have the N to insolence and micro-schema 2 Ny adj ↔ insolence, the emergence of schema 2 is based on the inheritance of information relating to Schema 1. Table 4 presents the attestation dates of Ny (schema 2) compared N (schema 1) based on OED data.

**Table 4.** Attestation dates of N and N-y with the sense 'insolent' in the OED (Smith 2021).

| Body Part Name | Other Figurative Senses | Schema 1 | Schema 2 |
|---|---|---|---|
| *Lip* OE | Yes | *Give lip* 1821 | *Lippy* 1875 colloq or dial |
| *Cheek* OE | No | *Have the cheek* 1823<br>*Give cheek* 1825 | *Cheeky* 1838 impudent |
| *Nerve* OE | Yes | *Have the nerve* 1887 | *Nervy* 1896 US colloq |
| *Face* 1300 | Yes | *Have the face* 1562 | *Facy* 1607 obs excep dial |
| *Front* 1290 | Yes | *Have the front* 1653 | *\*fronty* not attested |
| *Brow* OE | Yes | *Have brow* 1642 (obs) | *\*browy* not attested |
| *Chin* | Yes | *Have chin* 1877<br>*Give chin* 1877 | *Chinny* 1883 (sense talkative) |
| *Mouth* OE | Yes | *Be a mouth* 1699<br>*Be all mouth* (insolent)<br>*Give mouth* 1825 (express) | *Mouthy* 1589 (polysemic) |
| *Forehead* OE | Yes | *Have the forehead* 1564 (obs) | *\*foreheady* not attested |

Diachronic lexicographic data from the OED support this hypothesis that <*Have the N to*> construction may have led to the emergence of the morphological lexical construction, *Ny* adjective—insolence (*cheeky, nervy, gutsy, \*facy*) from the 1830s. *Have the cheek*, *have the nerve*, *have the guts* and *have the face* all predate the attestation of denominal adjectives *cheeky, nervy, gutsy* and *\*facy* (which has ultimately become obsolete).

Of course, the metaphorical or metonymical shift involved in the extension of the meaning of body part names to insolence no doubt has a deeper cognitive origin, since body parts and facial expressions are viewed as the manifestation (or origin) of interpersonal emotions. Body part names are well known to be particularly susceptible to semantic shifts (for instance, see Niemeier 2000; Hilpert 2006).

The existence of a number of competing *Have the N* expressions (both concept, emotion and body part nouns) shows that extensions are not blocked by synonymy relations (on the contrary, similarity is a factor in extension as argued in Goldberg 2019). Indeed, despite the frequency variations, some of these less frequent expressions may be ambivalent on the evaluation of the politeness–impoliteness spectrum. The ambivalence hinges on the indignation/disapproval versus approval rating, relating to the value attributed by the speaker to another person's actions or behavior. In (4), *nerve* takes a negative disapproval value, compared to (5) where *nerve* is associated with courage rather than insolence:

(4)   Ah, Mrs. Pampinelli. Mrs. Pampinelli. Look at her. All over the front page again. You know, she and I are really in the same business. I take pigs and turn 'em into sausages... and she takes our citizens and turns 'em into hams. Her last show was so bad . . . I didn't think she'd have the **nerve** to try and put on another one. Oh, she takes a lot of our young folks . . . and she turns their head by telling them they'd be great on the stage. (1935 *Doubting Thomas*, TV Movie)

(5)   That has been deliberate on my part. I better say this now or I won't have the **nerve** to say it later. (2001. *Sun Rising On the Hill District*, Robert Penny)

Although *guts* in (6) is interpreted as courage, most occurrences are compatible with a metapragmatic intention of daring or goading from the speaker as in (7). It is notable that many occurrences of *have the guts to* feature negative polarity as in (7) and (8).

(6)   But enlarged pictures of the individuals, separated from the total, disporting themselves in lewd, naked positions would do the job. Clearly the police must put a stop

to this. He would have every organization in the universe dedicated to dictating the morals of others on his side. No politician would have the **guts** to stand up in opposition. (1960 *Eight Keys to Eden*, Mark Clifton)

(7)  And you're not. Let's face it, kid. You don't have the **guts** to kill me. (1999. *Mystery Men*, TV Movie)

Despite the apparent synonymy of the patterns, these corpus extracts suggest that metapragmatic differences are present from one filler to the next. Behind their uses lie the question whether the ambivalence insolence/courage (disapproval/approval) is open, i.e., a matter of the content of the infinitival clause, or whether some fillers have a preferential negative or positive reading. Do *Have the N body part* constructions have a higher ambivalence reading than the *Have The N concept* constructions? In order to examine the question, we now turn to a collocate analysis in the COHA.

*3.3. Collocate Patterns in the Infinitival Clause*

A search for collocate patterns +1 to +5 is carried out to determine what types of verbs are significantly found in the *Have the N to* complement clause. Reducing the search to collocates in the +1 to +5 position does not target only verbs but all collocates found in this segment. The collocate search was carried out for specific *have the N to* structures for comparison: *heart, nerve, stomach, cheek, courage, temerity, audacity, impudence*. The results show a high frequency of collocates corresponding to speech act verbs and position taking verbs, such as *say, tell, speak, ask, admit, call, laugh* and *disappoint*.

The *raw* frequencies of collocates are compared to MI scores (Mutual Information) which provide a measure of the statistical likelihood of co-occurrence of the collocates with the target word (or its "proximity" see Goldberg 2019). This score therefore mitigates the frequency to provide some indication of the probability of co-occurrence. We will now compare collocate rankings based on a decreasing MI score. We should keep in mind that the collocates were not filtered to verbs only, although a large majority of hits correspond to verbs. We will not be discussing the noun collocates, many of which are pronouns (*me/themselves*) but have left the data in the tables for transparency.

We will proceed by analyzing several sets of synonymous patterns belonging to the insolence pragmeme; we begin with the body part filler *heart* versus the concept non filler *courage* representing positive evaluation (praise). We then compare three synonymous concept noun fillers (*impudence*, *audacity*, *temerity*), and finally, we compare two body part fillers representing negative evaluation (*gall* and *nerve*).

3.3.1. Collocates of Have the Heart and Courage

We turn *to have the heart* (1820–2010) and *have the courage* (1820–2010) which appear to be somewhat synonymous based on lexicographic information (OED); *Have the courage to* has 1194 tokens, *have the heart to* has 400 tokens. Note that this is slightly more than the figures provided in Table 1, but some variations were not taken into account in Table 1 due to thresholds for the general search for HAVE THE N TO.

Table 5 shows that top ranking MI scores correspond to more infrequent occurrences such as *reprimand*, *disappoint* and *scold*, whereas the more frequent collocates *ask* and *tell* result in much lower significance scores. This is expected due to the high overall frequency of general verbs such as *tell* and *ask*.

Similarly, Table 6 provides the list of collocates of *Have the courage* in a window of +1 to +1 to the right of the construction. These collocates are once again classified based on the MI score of the collocation.

The collocates of the pattern *have the courage* (which is almost three times more frequent than *have the heart*) follow a similar pattern. Frequent collocates such as the general verbs *tell*, *ask* and *speak* are not the highest ranking in terms of MI score. Instead, the more specific verbs *defy*, *confront* and *resist* find themselves at the top of the significance list, suggesting *have the courage* is followed by a verb indicative of disagreement/opposition (*resist, refuse, confront*) or risk taking (*risk, try*).

**Table 5.** Collocates of *Have the heart to* +1 to +5 in COHA.

| Collocates of *Have the Heart* | Frequency | Relative Freq | Mi Score |
|---|---|---|---|
| DISAPPOINT | 6 | 0.35 | 10.01 |
| REPRIMAND | 2 | 0.31 | 9.84 |
| SCOLD | 3 | 0.26 | 9.6 |
| ASSAIL | 2 | 0.24 | 9.47 |
| MEDITATED | 2 | 0.18 | 9.07 |
| DISTURB | 7 | 0.12 | 8.5 |
| REFUSE | 10 | 0.07 | 7.7 |
| SPOIL | 3 | 0.06 | 7.53 |
| REPROACH | 2 | 0.05 | 7.12 |
| COMMIT | 3 | 0.04 | 6.94 |
| UNDERTAKE | 2 | 0.04 | 6.81 |
| WAKE | 6 | 0.03 | 6.58 |
| DENY | 4 | 0.03 | 6.43 |
| ROB | 2 | 0.03 | 6.24 |
| TREAT | 3 | 0.02 | 5.88 |
| SHOOT | 4 | 0.02 | 5.78 |
| HARM | 3 | 0.02 | 5.76 |
| TELL | 52 | 0.02 | 5.74 |
| THROW | 5 | 0.02 | 5.62 |
| SUGGEST | 3 | 0.02 | 5.6 |
| DESERT | 3 | 0.02 | 5.56 |
| SUSPECT | 2 | 0.02 | 5.51 |
| WRITE | 6 | 0.01 | 5.26 |
| KILL | 6 | 0.01 | 5.14 |
| TURN | 12 | 0.01 | 5.09 |
| BLOW | 3 | 0.01 | 4.98 |
| BLAME | 2 | 0.01 | 4.96 |
| STRIKE | 3 | 0.01 | 4.84 |
| MENTION | 2 | 0.01 | 4.69 |
| ENJOY | 2 | 0.01 | 4.61 |
| BREAK | 4 | 0.01 | 4.57 |
| ASK | 8 | 0.01 | 4.51 |
| PICK | 2 | 0.01 | 4.17 |
| LAUGH | 2 | 0.01 | 4.02 |
| ENTER | 2 | 0.01 | 3.95 |
| DRIVE | 2 | 0 | 3.74 |
| SAY | 17 | 0 | 3.62 |
| EAT | 2 | 0 | 3.6 |
| SEND | 2 | 0 | 3.53 |
| STAND | 3 | 0 | 3.37 |
| LEAVE | 4 | 0 | 3.27 |
| STOP | 3 | 0 | 3.27 |
| AWAY | 10 | 0 | 3.2 |
| HIM | 47 | 0 | 3.17 |
| POOR | 3 | 0 | 3.05 |

As expected, based on the OED data for the noun *courage*, *have the courage* has a consistently positive value assessment as in (8) and (9) extracts, although this is mitigated by the larger (often dialogic) context:

(8) We want to know if you have the **courage** to print in your newspaper exactly what's wrong with San Francisco. That's right. I've waited a long time for this. (1935. *Barbary Coast*, TV Movie)

(9) What is it? I'm disobeying my own orders. You remember how I'd listen to the broadcasts? About all those people who've escaped? About that railroad train that broke through? I knew that was what you had in your mind. I've known it for a long time. But I never dreamed you'd ever have the **courage** to do it. (1953. *Man on a Tightrope*, TV Movie.)

**Table 6.** Collocates of *have the courage to* +1 to +5 in COHA.

| Collocates of *Have the Courage* | Frequency | Rel. Frequency | MI Score |
|---|---|---|---|
| DEFY | 8 | 0.31 | 8.29 |
| CONFRONT | 5 | 0.18 | 7.46 |
| RESIST | 9 | 0.08 | 6.39 |
| MISTAKES | 5 | 0.07 | 6.09 |
| CONFESS | 7 | 0.07 | 6.03 |
| REFUSE | 9 | 0.06 | 5.97 |
| ADMIT | 12 | 0.05 | 5.59 |
| SPEAK | 26 | 0.03 | 4.88 |
| ATTEMPT | 11 | 0.03 | 4.82 |
| ASK | 28 | 0.03 | 4.74 |
| STAND | 23 | 0.03 | 4.73 |
| OFFER | 9 | 0.02 | 4.52 |
| TELL | 58 | 0.02 | 4.32 |
| TRY | 18 | 0.02 | 4.24 |
| RISK | 5 | 0.02 | 4.24 |
| FIGHT | 11 | 0.02 | 4.23 |
| ENTER | 7 | 0.02 | 4.18 |
| KILL | 9 | 0.02 | 4.15 |
| CARRY | 8 | 0.02 | 4.1 |
| WRITE | 7 | 0.02 | 3.91 |
| ACT | 12 | 0.01 | 3.79 |
| THEMSELVES | 16 | 0.01 | 3.77 |
| FACE | 34 | 0.01 | 3.68 |
| DIE | 7 | 0.01 | 3.65 |
| SAY | 48 | 0.01 | 3.54 |
| LIVE | 12 | 0.01 | 3.54 |
| WALK | 7 | 0.01 | 3.53 |
| TRUST | 5 | 0.01 | 3.48 |
| REACH | 5 | 0.01 | 3.43 |
| TAKE | 35 | 0.01 | 3.19 |
| LEAVE | 11 | 0.01 | 3.15 |

### 3.3.2. Collocates of Have the Impudence, Audacity and Temerity

Based on the OED data for those entries, the French loan nouns *impudence*, *audacity* and *temerity* appear to be more compatible with a negative reading. The collocates of *have the impudence to* are shown in Table 7:

**Table 7.** Collocates of *have the impudence to* +1 to +5 in COHA.

| Collocates of *Have the Impudence* | Frequency | Rel Frequency | MI Score |
|---|---|---|---|
| WINK | 2 | 0.07 | 9.48 |
| REFER | 3 | 0.04 | 8.59 |
| GLARE | 2 | 0.03 | 8.5 |
| INSULT | 2 | 0.03 | 8.24 |
| MAINTAIN | 2 | 0.01 | 6.88 |
| CLAIM | 2 | 0.01 | 6.17 |
| ASK | 7 | 0.01 | 6.13 |
| SPEAK | 4 | 0 | 5.56 |
| CALL | 6 | 0 | 5.35 |
| SEND | 2 | 0 | 5.33 |
| PRESENT | 3 | 0 | 4.57 |
| TRY | 2 | 0 | 4.45 |
| TELL | 6 | 0 | 4.43 |
| ME | 24 | 0 | 4.1 |
| SAY | 6 | 0 | 3.93 |
| SET | 2 | 0 | 3.68 |
| FACE | 3 | 0 | 3.57 |

In the case of *have the impudence* (1820–1960), a similar pattern to *have the courage/heart* emerges with verbs of speech such as *tell*, *call* and *speak*, having higher frequencies and lower MI scores. At the top of the MI ranking are *wink*, *refer*, *glare* and *insult*, more specific verbs implying some risk-taking and boundary-pushing on the part of the co-speaker.

Corpus examples for *impudence* show a high level of heightened negative assessment in (10) and (11), which is the most recent occurrence in the COHA.

(10) "Miss!" roared the old man, bringing down his cane with a resounding thump upon the floor; "miss! how dare you have the impudence to face me, much less the – the – the **assurance!** – the **effrontery!** – the **audacity!** – the **brass**! to speak to me!" (1867. *Hidden Hand*. Southworth, Emma.)

(11) I am surprised that after your insolent references to myself, Sir Osbert and Mr. Sacheverell Sitwell her younger brother made in verse some years ago, you should have the **impudence** to invite me to waste my time at your show. (*Time Magazine*: 1961/10/12.)

Tables 8 and 9 representing *have the temerity* (1830–2010) and *have the audacity* (dates) confirm the larger pattern of collocates representing conflict and disagreement. (12) shows the righteous indignation implied in the expression, which appears to be being phased out based on diachronic occurrences in the COHA.

**Table 8.** Collocates of *have the temerity to* +1 to+5 in COHA.

| Collocates of *Have the Temerity* | Frequency | Rel Frequency | MI Score |
|---|---|---|---|
| DISAGREE | 2 | 0.07 | 9.13 |
| EXECUTE | 2 | 0.06 | 8.83 |
| SUGGEST | 9 | 0.05 | 8.51 |
| CHALLENGE | 4 | 0.03 | 7.66 |
| DECLARE | 2 | 0.02 | 7.31 |
| VENTURE | 2 | 0.02 | 7.24 |
| MAINTAIN | 3 | 0.02 | 6.99 |
| ATTEMPT | 6 | 0.02 | 6.85 |
| DENY | 2 | 0.01 | 6.75 |
| APPROACH | 3 | 0.01 | 6.3 |
| EXPRESS | 2 | 0.01 | 6.1 |
| ADDRESS | 2 | 0.01 | 5.85 |
| WRITE | 3 | 0.01 | 5.59 |
| DIRECTLY | 2 | 0.01 | 5.46 |
| ASK | 6 | 0.01 | 5.42 |
| LAUGH | 2 | 0.01 | 5.34 |
| ENTER | 2 | 0.01 | 5.28 |
| QUESTION | 3 | 0 | 4.21 |

(12) And you mean to tell me the Germans have the **temerity** to attempt a raid in the very mouth of the Thames?" Lord Hastings nodded. "They certainly have," he said quietly. (1915. *The Boy Allies Under the Sea*, Robert Drake.)

(13) is the most recent occurrence in the COHA, and illustrates the strong domineering judgment implied by this more outdated expression.

(13) How dare those state workers take it in the shorts for a full career of service to the public and then have the **temerity** to expect a decent pension? J. Brandeis Sperandeo, Denver (2010. *Denver Post Open Forum*)

**Table 9.** Collocates of *have the audacity to* +1 to +5 in COHA.

| Collocates of *Have the Audacity* | Frequency | Rel Frequency | MI Score |
|---|---|---|---|
| INSULT | 4 | 0.06 | 8.22 |
| PROPOSE | 4 | 0.05 | 7.91 |
| ASSERT | 2 | 0.04 | 7.68 |
| INVITE | 2 | 0.03 | 7.24 |
| ADOPT | 2 | 0.03 | 7.05 |
| DEMAND | 7 | 0.02 | 6.8 |
| LAUGH | 7 | 0.02 | 6.61 |
| PAYMENT | 2 | 0.02 | 6.52 |
| DENY | 2 | 0.01 | 6.22 |
| CLAIM | 3 | 0.01 | 5.74 |
| ATTEMPT | 4 | 0.01 | 5.73 |
| ADDRESS | 2 | 0.01 | 5.31 |
| RAISE | 2 | 0.01 | 5.31 |
| CITIZENS | 2 | 0.01 | 5.19 |
| ASK | 7 | 0.01 | 5.11 |
| STRIKE | 2 | 0.01 | 5.05 |
| ESCAPE | 2 | 0.01 | 4.95 |
| SIT | 3 | 0 | 4.66 |
| EXPECT | 2 | 0 | 4.62 |
| DRIVE | 2 | 0 | 4.53 |
| CALL | 6 | 0 | 4.33 |
| SAY | 16 | 0 | 4.32 |
| TELL | 10 | 0 | 4.15 |
| TRY | 3 | 0 | 4.02 |
| HOLD | 3 | 0 | 3.98 |
| STAND | 2 | 0 | 3.57 |
| CHURCH | 2 | 0 | 3.55 |
| BRING | 2 | 0 | 3.44 |
| THINK | 9 | 0 | 3.4 |
| FIRE | 2 | 0 | 3.33 |
| TURN | 2 | 0 | 3.29 |
| RUN | 2 | 0 | 3.29 |
| USE | 3 | 0 | 3.21 |

*Have the audacity* to (1830–2010) is still in use, with a stronger pattern of usage until 2010 than *have the temerity* which appears to be phased out, becoming archaic. In (15), the extract taken from another forum is also suggestive of sarcasm or irony (see Garmendia 2018; Lehmann 2023), defined as an incongruence between content and opinion.

(14) But a true word, fresh from the lips of a true man, is worth paying for, at the rate of eight dollars a day, or even of fifty dollars a lecture. The taunt must be an outbreak of jealousy against the renowned authors who have the **audacity** to be also orators. (1859. *Autocrat of the Breakfast Table*, Oliver Holme)

(15) don't get why they are so incensed that American citizens would have the **audacity** to want to know the truth about the war in Iraq. It is our right in a democracy to question and demand the truth from our leaders. It is dangerous not to question and demand. (2005. *San Francisco Chronicle: Letters to the editor.*)

3.3.3. Collocates of Have the Gall and Nerve

*Have the gall* (1910–2010) and *Have the nerve* (1830–2010) appear semantically similar; however, the ambivalence of *nerve* is not present with *gall*, which appears lexically far more on a derogatory scale. The collocates of *gall* in Table 10 however do not clearly show negative polarity, with a similar collocate list to the other patterns.

**Table 10.** Collocates of *have the gall* +1 to +5 in COHA.

| Collocates of *Have the Gall* | Frequency | Rel Frequency | MI Score |
|---|---|---|---|
| ME | 11 | 0 | 3.09 |
| SAY | 7 | 0 | 4.27 |
| COME | 7 | 0 | 3.97 |
| TELL | 6 | 0 | 4.55 |
| ACTUALLY | 3 | 0.01 | 5.86 |
| ASK | 3 | 0 | 5.02 |
| CALL | 3 | 0 | 4.47 |
| ACCUSE | 2 | 0.08 | 9.83 |
| CLAIM | 2 | 0.01 | 6.29 |
| THROW | 2 | 0.01 | 6.23 |
| WALK | 2 | 0 | 5.23 |
| STAND | 2 | 0 | 4.71 |
| SPEAK | 2 | 0 | 4.68 |
| TRY | 2 | 0 | 4.57 |
| SELF-DEFENSE. | 1 | 4.35 | 15.59 |
| HELIOGRAPH | 1 | 3.85 | 15.41 |
| DISENFRANCHISE | 1 | 3.7 | 15.35 |
| RISIBLE | 1 | 1.89 | 14.38 |
| OBJETS | 1 | 0.81 | 13.17 |
| HUFFY | 1 | 0.63 | 12.8 |
| SAUNTER | 1 | 0.33 | 11.85 |
| POTTERS | 1 | 0.25 | 11.48 |
| CONFISCATE | 1 | 0.25 | 11.47 |
| PLATITUDES | 1 | 0.2 | 11.12 |
| UNINVITED | 1 | 0.19 | 11.05 |
| SMUGGLE | 1 | 0.18 | 10.95 |
| DEPLORE | 1 | 0.11 | 10.28 |

The occurrence in 16 shows high levels of animosity in the form of a rhetorical question:

(16)  The other flushed a deep red. "Until then," said Barney easily, "I shall be forced to regard you and your officers as prisoners of war. "Johns laid his hand on his sword at this Yankee impudence. "You have the **gall** to let the matter go unreported all day? And to hold me aboard?" (1951. *Captain Barney: A novel*. Jan Westcott)

(17)  When Susan Weiner first ran for mayor here five years ago, at least one local was appalled that a New York Republican newcomer would have the **gall** to think she could just waltz in and take over this genteel old Southern city. (1995. *Atlanta Journal Constitution* Our Southern Yankee, Jingle Davis)

(16) and (17) show that the type of verb in the infinitival complement slot need not be particularly damning for a derogatory interpretation to emerge. The triggers of indignation lie in the metapragmatic context between the speakers dictated by their socio-cultural positions and functions.

*Have the nerve* (1830–2010) on the other hand is both more frequent and less polarizing than *have the gall*. The collocates ranked according to their MI score in Table 11 show, however, that the construction remains somewhat predictable.

**Table 11.** Collocates of *Have the nerve* +1 to +5 in COHA.

| Collocates of *Have the Nerve* | Frequency | Rel Frequency | MI Score |
|---|---|---|---|
| HOUSEWIVES | 2 | 0.14 | 8.49 |
| CONTRADICT | 2 | 0.14 | 8.47 |
| TACKLE | 3 | 0.09 | 7.74 |
| UNDERTAKE | 2 | 0.04 | 6.56 |
| ASK | 36 | 0.03 | 6.44 |
| INVITE | 2 | 0.03 | 6.21 |
| JUMP | 4 | 0.03 | 6.14 |
| DRAG | 2 | 0.03 | 5.98 |
| PROPOSITION | 2 | 0.02 | 5.6 |
| EXAMINE | 2 | 0.02 | 5.57 |
| RESIST | 2 | 0.02 | 5.55 |
| TRY | 17 | 0.02 | 5.49 |
| BOTHER | 2 | 0.02 | 5.36 |
| REFUSE | 2 | 0.01 | 5.13 |
| SHOOT | 3 | 0.01 | 5.12 |
| PUSH | 2 | 0.01 | 4.94 |
| SUGGEST | 2 | 0.01 | 4.77 |
| WRITE | 5 | 0.01 | 4.75 |
| CALL | 16 | 0.01 | 4.72 |
| STAND | 9 | 0.01 | 4.71 |
| WEAR | 3 | 0.01 | 4.68 |
| TELL | 29 | 0.01 | 4.65 |
| THROW | 3 | 0.01 | 4.64 |
| KISS | 2 | 0.01 | 4.49 |
| STICK | 2 | 0.01 | 4.44 |
| CHARGE | 4 | 0.01 | 4.41 |
| CLAIM | 2 | 0.01 | 4.12 |
| BRING | 6 | 0.01 | 3.99 |
| RIDE | 2 | 0.01 | 3.99 |
| PICK | 2 | 0.01 | 3.93 |
| KILL | 3 | 0.01 | 3.9 |
| ANYONE | 3 | 0.01 | 3.87 |
| START | 4 | 0.01 | 3.79 |
| LAUGH | 2 | 0.01 | 3.77 |
| DIE | 3 | 0.01 | 3.76 |
| ATTEMPT | 2 | 0.01 | 3.7 |
| WALK | 3 | 0 | 3.64 |
| SAY | 20 | 0 | 3.61 |
| PUT | 12 | 0 | 3.55 |
| ACT | 4 | 0 | 3.54 |
| GO | 24 | 0 | 3.47 |
| CARRY | 2 | 0 | 3.43 |
| BUY | 2 | 0 | 3.38 |
| RUN | 4 | 0 | 3.25 |
| COME | 17 | 0 | 3.08 |
| ME | 49 | 0 | 3.07 |

In the case of *have the nerve,* the collocate *ask* is ranked higher on the MI score than other verbs of speech, *tell, claim* and *call*, indicating a higher attractivity for the slot than in the case of have the N (concept). The occurrences of *ask* show the ambivalence of *have the nerve*. Compare the positive evaluation in (18) with (19) where *have the nerve* is indicative of disbelief and negative evaluation:

(18)　Of course I had to go, after that – and I nearly killed myself. I thought I was pretty good to even try it. Nobody else in the party tried it. Well, afterward Rosalind had the **nerve** to ask me why I stooped over when I dove.' It didn't make it any easier,' she said,' it just took all the courage out of it. (1920. *This Side of Paradise.* Scott Fitzgerald.)

(19) That does it, I've had it." Harry stilled. "What's wrong?" "You have the **nerve** to ask me what's wrong? After telling me that you're not going to approve Duncan's proposal?" (1997. *Absolutely, positively,* Krentz, Jayne Ann.)

The other verbs in the infinitival position that are ranked higher are *contradict* and *tackle*, verbs of speech acts indicative of conflict or disagreement. In the case of (20), they are indicative of a sense of entitlement or high regard for one's ability (self-assurance).

(20) Skeptics who include other law enforcers have had the **nerve** to contradict Hoover on the point, and to suggest that the crimes record division of the FBI is also its public relations operation. (1970 *The New Republic*: 11/28/70.)

(21) But later on I may tip you off to a lot of things that Morrow did that a Diplomat didn't. do. In fact, he told me one time that he didn't believe he would have had the **nerve** to tackle the job, if it. was not that he had the example of Diplomats to watch, so that he could do the opposite and feel sure of success. (1932. *Saturday Evening Post: Letters of a Self-Made Diplomat to Senator Borah*)

The success of the *have the nerve* pattern historically may be accounted for by the ambivalence in its usage, compared to other patterns. We now take a closer look at body part fillers and their usage profiles in the COHA.

*3.4. Diachronic Variation in Fillers: Density*

3.4.1. Concept Noun Fillers over Time

As we have observed, concept noun fillers are far more frequent than body part fillers. This is expected given the wide range of concept nouns available, and the much more limited availability of body part nouns. The frequency per mil of *Have the N to* appears to be relatively stable over the COHA period as shown in Figure 3.

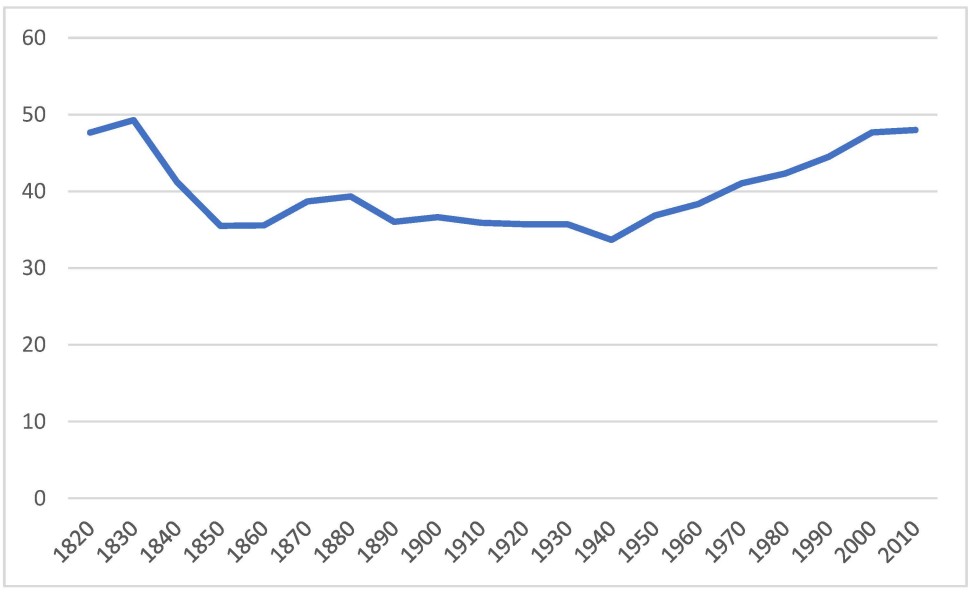

**Figure 3.** Relative frequency per decade of HAVE THE N TO in COHA (1820–2010).

This means the pattern itself is likely established by 1810 based on concept nouns of attitude and emotion, with the development of body part fillers from the 1800s. This can be verified based on OED attestations for the senses of body parts as shown in Smith (2021). However, we can question how the usage of filler nouns in the COHA has evolved over the course of the 200 year span of the corpus.

As is well known, one of the main issues in tracking change is establishing if variation can be interpreted as shift, and if some directionality can be established, for instance towards higher intersubjective readings. In the case of *have the N to* patterns, collocates appear remarkably similar with speech act verbs being prevalent. What distinguishes the

patterns are specific metapragmatic readings and the level of ambivalence of readings. More than frequency itself, we can argue it is the prototypicality of the association of a pattern and a reading that indicates the strength of the pattern, therefore, its onomasiological productivity.

The capacity for innovation of *have the N to* is inherently limited to the class of filler nouns of concepts and body parts relating to politeness and impoliteness but within that, field variation is present and extensibility is possible although not broad.

### 3.4.2. Body Part Fillers over Time

In this subsection, we look at diachronic changes in frequency starting with body part nouns. A closer look at the frequencies and dates of attestation in the COHA for *have the N (body part)* reveals higher frequencies than found in Section 3.1 based on the queries *Have the N to*. There is no specific search query enabling us to select body part nouns in the complement position N. A manual search for each fully instantiated lemmatized expression *Have the N to* provided seven expressions: *Have the nerve to* (473), *have the guts to* (305), *have the gall to* (105), *have the face* (71), *have the cheek to* (18), *have the stomach to* (15), *have the balls to* (58). These results shown in Table 12 include all verbal forms of *have* (*having, 've, had*, etc.).

**Table 12.** Compared frequency of body part fillers *have the N to*.

| Lemmatized Expression | Tokens |
|:---:|:---:|
| *have the guts to* | 305 |
| *have the stomach to* | 15 |
| *have the nerve to* | 473 |
| *have the gall to* | 105 |
| *have the cheek to* | 18 |
| *have the face to* | 71 |
| *have the balls to* | 58 |

Figure 4 shows the diachronic distribution of the tokens for each of the seven patterns in the COHA, providing a timeline of their compared usage in the corpus over time.

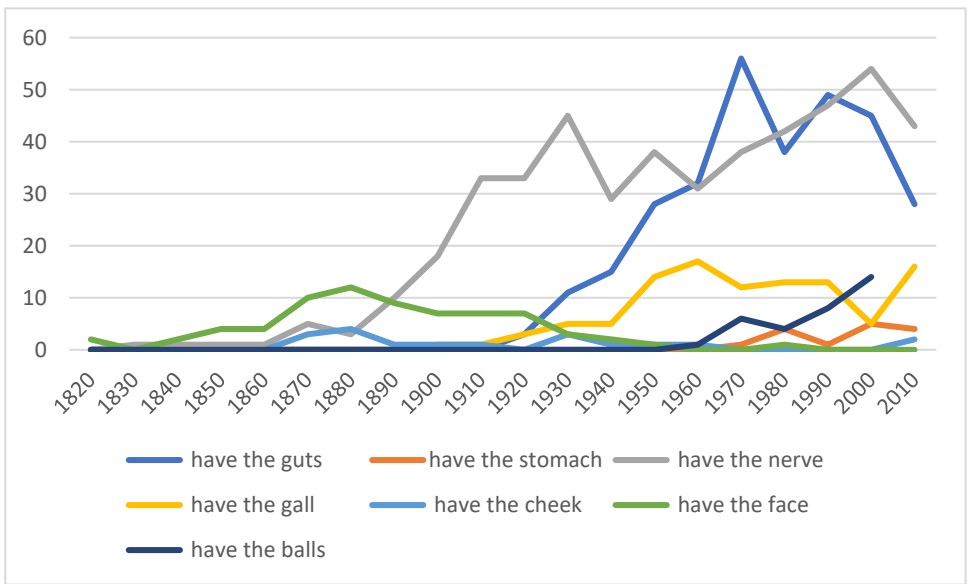

**Figure 4.** Frequency evolution of seven *Have the N* (body part) *to* patterns in COHA.

Of the seven patterns, *have the face* emerged earliest and has died out from the 1940s. *Have the cheek* is attested earlier than *have the stomach*, which appears to be a recent extension (1970). *Have the nerve* has the strongest progression from the 1880s, and *have the guts* has increased steadily from the 1930s until the present. *Have the stomach* emerges latest in the

1970s, with *have the balls* emerging around the same period. The first occurrence of *have the balls* [1969] is situated in a sexually explicit fictional dialogue in (22).

(22)   Nobody will see you, Darrell says. It's too cold for doctors or nurses to be out and nobody else cares. And if you actually do have the **balls** to fuck her you get to be leader of our club for as long as you're here, Corky says as the others solemnly nod their approval. That's true, Timmy, Darrell says. That's how Big Jim got to be boss and that's how I got to be second-in-command. They really do take me for an idiot, I think to myself. (1969. *Listen to the Silence*, David W. Elliot)

Most of the subsequent occurrences are from TV or movie sources exhibiting a high level of oral colloquialism and tolerance for vulgar language (with other references to dysphemistic abuse terms).

(23)   Oh, really? "Marcus the Magnificent." The boy, the genius, the legend. That's their name, okay? Not mine. I'm willing to bet the only reason you came to Burrows in the first place, and not one of those Ivy League schools, is because you didn't have the **balls** to measure yourself against real competition. You're a phony. A fake. Thank you for your time, Professor, and your, uh, constructive criticism. (2018. *The Truth About the Harry Quebert Affair*, TV Movie)

At the opposite end of the scale, *have the cheek* appears to have the lowest altogether frequency of usage despite a longer lifespan (1870–). (24) illustrates one of the earliest occurrences, and (25) one of the later occurrences which have become scarce in the COHA.

(24)   With all the channels of truthful information thus open and unobstructed, you preferred to get what you wanted from a spy. Mr. Howard has the **cheek** to proclaim that during the "labors" of his committee, instead of acting upon honest and legitimate evidence, he sent inquiries to this secret informer, who answered by giving information of "great importance," but his communications "were always indirect and anonymous!" (1871. *Galaxy* February 1871: 257–276; *Mr Black to Mr Wilson*. J.S. Black.)

(25)   While he was getting undressed last night she'd had the **cheek** to say, "Those pants are getting tired". (2011. *Night Thoughts*. Helen Simpson)

From this, it could be suggested that *have the cheek* is on a path of regression being outperformed by more recent and more dysphemistic terms such as *stomach* and *balls*. However, the issue of the variety of English is certainly in question, as is the type of data provided by a specific corpus. A search for *have the cheek* in the contemporary OEC corpus or Oxford English Corpus (2000–2010), via Sketch Engine, shows *have the cheek* to have a non-negligible relative frequency of 0.04 per mil. (26) is taken from a rather scathing literary review with a tone of strong condescension and disapproval:

(26)   With the exception of Amy Sohn and Benjamin Anastas (the author of the cult novel, An Underachiever's Diary) and his terrific essay here, "An Unexamined Life", most of the writers in this anthology have the **cheek** to pronounce Salinger 's prose variously as "failed poetry", "brilliant writing steam rolling everything", "original without being good", "workmanlike", relentlessly middleclass and middlebrow—while their own prose is artless, ungenerous, mediocre. (May 2002. *The Hindu: Literary Review*.)

3.4.3. Issues with Low Frequency

Low frequencies raise a number of methodological issues in corpus linguistics. In addition to the issue of the data pool, raw frequency patterns can also be deceptive in that they rely on the assumption that individual patterns are equivalent or at least comparable. However, a collocate analysis has shown that these expressions are not equally representative of the insolence pragmeme (although it must be said that the accuracy of a collocate analysis also depends on the availability of data).

In any case, from a semantic pragmatic standpoint, it is important to note that these variations of the pattern are not all synonymous and represent a metapragmatic comment

on attitude which ranges from positive evaluation (courage) to negative evaluation (impudence). The readings of the evaluations are, however, not readily available outside of any context, although some tend to be more typically associated with negative evaluation; the ambivalence of the reading appears to be part of the construction itself. The collocate analysis provides some qualitative clues as to the specific usage of the individual patterns, and their specific inferences. Ambivalence appears to be a factor in the differentiation of the patterns. *Have the nerve* is notably extremely variable in terms of value assigned to the metapragmatic comment, which ranges from laudatory to accusatory.

On the other hand, the surprisingly low-key *have the cheek* appears deceptively insignificant in the graph and the data set. However, it can be argued that *have the cheek* is a well-entrenched formulaic expression despite its relative infrequency compared to other fillers. The entrenchment value, or prototypical value of the expression in the sense of 'insolent', is a measure that remains difficult to assess purely from a usage-based perspective. If the consistency of the semantic and pragmatic reading has an effect on the representativeness of that expression within the conceptual space, then, despite its low frequency, *cheek* (and *cheekiness)* is likely a central representative of a certain subtype of insolence characterized by a positive evaluation as shown in Smith (2021).

The signs of extensibility of *have the N* (body part) are present, although not major with three new body part constructions appearing in the 1900s including *guts*, *stomach* and *balls*. The extension of the pattern to more dysphemistic body part names is likely related to the greater proportion of colloquial oral discourse in recent corpora. We may note that the emergence of *have the stones* and *have the cojones* is likely triggered via synonymy with *balls*. *Stones* has an extremely low frequency rate in the COHA with a total of 3 occurrences ranging from 1980 to 2010; *cojones* appears once in the COHA in a TV/Movie dialogue line from [1991] in (27):

(27) Now all you have to do is have the **cojones** to say it to your boss instead of his secretary. (1991. *Suburban Commandon*, TV Movie)

Yet again this low frequency cannot be equated with low productivity but rather is a sign of innovation. The much larger OEC gives a relative frequency of 0.02 for *have the stones* (28) showing that it is far from insignificant; *have the cojones* has a frequency of 0.01.

(28) In all honesty, I did not think that Jack McConnell and his cohorts would have the **cojones** to take on Scotland's suicidal, macho fag-and-booze culture, so congratulations to them (5 February 2006. *Scotland on Sunday*.)

We believe that these remarks and issues point to the need to pursue in an depth low-level study of micro constructions from a comparative perspective.

## 4. Discussion of Onomasiological Productivity and "Conceptual Space"

We now discuss our findings concerning HAVE THE N (BODY PART/ATTITUDE) TO ↔METAPRAGMATIC COMMENT, specifically regarding the issue of how to overcome methodological issues relating to a qualitative approach to the constructional architecture and the relative productivity of constructions. The corpus-driven methodology attempted to study a low-level construction belonging to a higher-level pragmatic construction (insolence/(im)politeness) and represents a metapragmatic discourse comment on a co-speaker's behavior or attitude.

The analysis of the occurrences of the pattern in the COHA showed a high frequency of certain concept words in structures that have become phraseological patterns (*have the right to*), resembling modal comments.

Our question was whether an onomasiological viewpoint would be able to assess the extensibility capacity of a construction; the study has shown that extensibility is present in terms of the inclusion of new body part nouns to the pattern in the 1970s such as *guts*, *stomach* and *balls* which follow the metaphorical pathway of BODY PART IS COURAGE or BODY PART-IS INSOLENCE. However, beyond this observation, it is difficult to assess what proportion of the onomasiological space these constructions take. The method applied

here showed the intricacies of identifying patterns with respect to semantic, pragmatic and conceptual criteria, and also revealed its limitations. We used raw frequencies, relative frequencies and collocate behaviors to trace *have the N to* patterns over time in the COHA. The wide variety of fillers in the N position were categorized into broad conceptual categories (emotions and attitudes) and body part noun fillers were compared to conceptual noun fillers. The tools used included raw frequencies, relative frequencies and collocate patterns to establish a preliminary view of the diachronic evolution of *have the N to*. The results show that what is required is a more cohesive picture of the onomasiological landscape of the politeness metapragmatic comment superconstruction.

We argue that the results confirm that frequency parameters alone do not adequately show the significance of a pattern on a cognitive level, as is illustrated by the poor data for *have the cheek*, despite the high entrenchment (that is the high degree of mental correlation between cheek–insolence) of this pattern in the lexicon of speakers. We believe this shows the importance of continuing this line of reasoning and testing a larger number of subconstructions within a network of superconstructions. If we follow Goldberg (2019, p. 67), who underlines that studies have shown that "clusters with higher density tended to attract new members, just as the notion of coverage predicts", then the density of body part fillers associated with insolence/(im)politeness points to a high coverage of conceptual space.

Although this study focused on a very low-level construction, we argue that studying more low-level lexico-grammatical patterns can provide a more reliable understanding of the emergence and development of a family of constructions. If we look at grammaticalization and constructional change, increasing intersubjectivity (expression of approval or disapproval and indignation) can be expressed by low-level lexicogrammatical patterns which in turn point to the relevance of carrying out corpus-driven studies of expressions that are not fully grammaticalized. The expression of modality and intersubjectivity is far from being the purview of grammar only, which therefore continues to call into question the nature of the lexicon-grammar interface. We can only concur with Goldberg's (2019, p. 146) desire "to capture relational meanings or the myriad ways in which words and grammatical constructions combine to give rise to interpretations in context". To contribute to achieving this goal, a more in-depth study of collocational behavior in diachronic corpora (including corpora representing periods preceding 1810, such as the English Historical Book Collection or EHBC) is required. Furthermore, the identification of the family of constructions belonging to the insolence pragmeme will also provide a more accurate assessment of shift and productivity from the onomasiological or conceptual standpoint. Ultimately, we aim to achieve a better understanding of the organization of the constructional families and the nature of the constructional families.

**Funding:** This research received no external funding.

**Institutional Review Board Statement:** Not applicable.

**Informed Consent Statement:** Not applicable.

**Data Availability Statement:** All data is freely available using the COHA.

**Acknowledgments:** Many thanks to the expert reviewers and editing team who helped to considerably improve on earlier versions of this paper. All remaining errors are mine entirely.

**Conflicts of Interest:** The author declares no conflict of interest.

## Notes

[1]  "Any linguistic pattern is recognized as a construction as long as some aspect of its form or function is not strictly predictable from its component parts or from other constructions recognized to exist. In addition, patterns are stored as constructions even if they are fully predictable as long as they occur with sufficient frequency." (Goldberg 2006, p. 5) Note, however, that Goldberg (2019) has since criticized this reference to "sufficient frequency" as "nonsensical".

[2]  "Changes in form and meaning of a construction can be studied through frequency measurements of its variants: An important concept in the present work is thus the idea that constructions are not fixed, but flexible, displaying formal and functional variation."(Hilpert 2013, p. 6).

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
