# Peer review of "Productivity from a Metapragmatic Perspective: Measuring the Diachronic Coverage of the Low Level Lexico-Grammatical Construction Have the N (Body Part/Attitude) to ↔<Metapragmatic Comment> Using the COHA"

_languages, doi:10.3390/languages8020092_

Round 1

Reviewer 1 Report

This paper takes a highly promising approach to a very interesting topic, or rather: a number of very interesting topics (the have the N construction, productivity, coverage...). While I am very sympathetic to the overall thrust of the paper, I think it needs a fairly thorough revision to be publishable. In its current form, it is, metaphorically speaking, more of a raw diamond that presents interesting data and some relatively vague ideas of how they can be interpreted with regard to the key topics mentioned in the title of the paper, but they partly remain a bit underdeveloped, and the connection between them could become clearer. In particular, I see the following issues:

- The paper seems pretty hastily written, which is understandable given the rather low-key publication venue but is still detrimental to the paper's reader-friendliness. Typos and grammatical errors abound, but that can be easily remedied with some proofreading. All tables are simply screenshots, either from Excel or from the english-corpora.org interface, which is particularly problematic in Table 2, where the last digit of each year is placed after the line wrap. Also, the bars in the last row (Table 2) or column (e.g. Table 5) remain unexplained. I would suggest to insert the tables as "real" tables, rather than screenshots, and to only include the numbers that are actually relevant for the paper.

- In Table 1, concept nouns and body part nouns are distinguished. These categories are never really introduced, and it remains unclear why the former are subsumed under the maximally vague notion of "concept".

- Other methodological choices should be justified in more detail as well. For instance, why is the collocation analysis limited to heart and courage (and a few other nouns)? Given the constructionist approach taken in the paper, it might be interesting to take all noun-verb combinations into account and use a method like e.g. covarying-collexeme analysis. This would of course require some manual work, and I'm not sure if it's actually possible given the limitations of the english-corpora.org interface, but if at all possible, it would be great to work with an exhaustive list of N and V fillers... Otherwise, the limitations of the english-corpora interface should just be made more transparent.

- The discussion in Section 4 is relatively short; the key concepts mentioned in the title - "productivity" and "coverage" - could be discussed in more detail. Especially given the rich literature on different concepts of productivity, it might be worthwhile to say a few words about the productivity concept underlying the case study in the paper, and how exactly "productivity" and "coverage" are related, and how the results can inform models of coverage and productivity. Section 4 does say a few things about these topics but still remains relatively vague.

All in all, I think this is a paper with high potential but it doesn't really live up to what it promises; however, a thorough revision could turn it into a really good paper.

Reviewer 2 Report

- Please check for consistent use of British or American English spelling. 

- Is it possible that the data will be made available with the publication of the article? 

All further comments can be found directly in the attached pdf-file.

Round 2

Reviewer 1 Report

The author/s has/have done an impressive job in addressing the comments from the previous reviewing round, rewriting large parts of the paper. The line of argument is much clearer now, and I think the paper can be published in its present form

Author Response

I would like to thank both reviewers sincerely for their detailed comments and suggestions which have helped me improve the first draft of this paper. I am very grateful for the constructive criticism and feedback.